

# Ten years of progress and promise of induced pluripotent stem cells: historical origins, characteristics, mechanisms, limitations, and potential applications

Adekunle Ebenezer Omole[1] and Adegbenro Omotuyi John Fakoya[2]

[1] Department of Basic Sciences, American University of Antigua College of Medicine, St. John's, Antigua
[2] Department of Anatomical Sciences, All Saints University, School of Medicine, Roseau, Dominica

Corresponding authors
Adekunle Ebenezer Omole,
kunlesty@yahoo.com
Adegbenro Omotuyi John Fakoya,
gbenrofakoya@gmail.com

## ABSTRACT

The discovery of induced pluripotent stem cells (iPSCs) by Shinya Yamanaka in 2006 was heralded as a major breakthrough of the decade in stem cell research. The ability to reprogram human somatic cells to a pluripotent embryonic stem cell-like state through the ectopic expression of a combination of embryonic transcription factors was greeted with great excitement by scientists and bioethicists. The reprogramming technology offers the opportunity to generate patient-specific stem cells for modeling human diseases, drug development and screening, and individualized regenerative cell therapy. However, fundamental questions have been raised regarding the molecular mechanism of iPSCs generation, a process still poorly understood by scientists. The efficiency of reprogramming of iPSCs remains low due to the effect of various barriers to reprogramming. There is also the risk of chromosomal instability and oncogenic transformation associated with the use of viral vectors, such as retrovirus and lentivirus, which deliver the reprogramming transcription factors by integration in the host cell genome. These challenges can hinder the therapeutic prospects and promise of iPSCs and their clinical applications. Consequently, extensive studies have been done to elucidate the molecular mechanism of reprogramming and novel strategies have been identified which help to improve the efficiency of reprogramming methods and overcome the safety concerns linked with iPSC generation. Distinct barriers and enhancers of reprogramming have been elucidated, and non-integrating reprogramming methods have been reported. Here, we summarize the progress and the recent advances that have been made over the last 10 years in the iPSC field, with emphasis on the molecular mechanism of reprogramming, strategies to improve the efficiency of reprogramming, characteristics and limitations of iPSCs, and the progress made in the applications of iPSCs in the field of disease modelling, drug discovery and regenerative medicine. Additionally, this study appraises the role of genomic editing technology in the generation of healthy iPSCs.

## INTRODUCTION

The development of induced pluripotent stem cells (iPSCs) in 2006 by Shinya Yamanaka was a remarkable breakthrough that was made possible by many research findings by past and current scientists in related fields. In 1962, Sir John Gurdon achieved the first example of cellular reprogramming by reporting the generation of tadpoles from enucleated unfertilized frog egg cells that had been transplanted with the nucleus from intestinal epithelial somatic cells of tadpoles (*Gurdon, 1962*). This remarkable method of reprogramming somatic cells to the pluripotent embryonic state with the same genetic makeup was termed somatic cell nuclear transfer (SCNT). This discovery led to the birth of cloning. Thirty-five years later, Sir Ian Wilmut and his team used the same SCNT strategy of cellular reprogramming in the cloning of Dolly the sheep, the first mammalian to be generated by somatic cloning (*Wilmut et al., 1997*). These two scientific breakthroughs in somatic cloning proved that the nuclei of differentiated somatic cells contain all the necessary genetic information to generate a whole organism and that the egg cell contains the necessary factors to bring about such reprogramming. In 2001, *Tada et al. (2001)* further lent credence to the somatic cloning hypothesis through another novel strategy of reprogramming termed cell fusion. The cell fusion of somatic cells with embryonic stem cells (ESCs) to generate cells capable of expressing pluripotency-related genes showed that ESCs do contain some factors that can reprogram somatic cells (*Tada et al., 2001*). There are two other important landmarks—the generation of mouse ESCs cell lines in 1981 by Sir Martin Evans, Matthew Kaufman and Gail R. Martin and the subsequent generation of human ESCs in 1998 by James Thomson (*Evans & Kaufman, 1981*; *Martin, 1981*; *Thomson et al., 1998*). The ESCs are developed from pre-implantation embryos and are capable of generating any cell type in the body; an inherent characteristic termed pluripotency. Their discoveries shed light on the appropriate culture conditions and transcription factors necessary for the maintenance of pluripotency. The merging of all these essential historical landmarks led to the discovery of iPSCs (Fig. 1).

But why the need for iPSCs since they are pluripotent just like ESCs? Firstly, the use of ESCs is fraught with strong ethical concerns related to embryo destruction, and this has hindered its clinical application. Secondly, there are the safety concerns related to the immune rejection of the ESCs. Finally, due to its source from the embryo, ESCs are limited in supply, and this will limit broader therapeutic application. Hence, there was an urgent need for another substitute for ESCs that bypasses these important drawbacks. Indeed, the iPSCs serve as an alternative source of pluripotent stem cells with the same differentiation potential as ESCs while avoiding the ethical issues associated with the latter.

Shinya Yamanaka and Kazutoshi Takahashi developed the mouse iPSCs in 2006 through a different method of reprogramming: the use of a retrovirus to deliver into a somatic cell (mouse fibroblast), a combination of four reprogramming transcription factors, including Oct 3/4 (Octamer-binding transcription factor-3/4), Sox2 (Sex-determining region Y)-box 2, Klf4 (Kruppel Like Factor-4), and c-Myc nicknamed
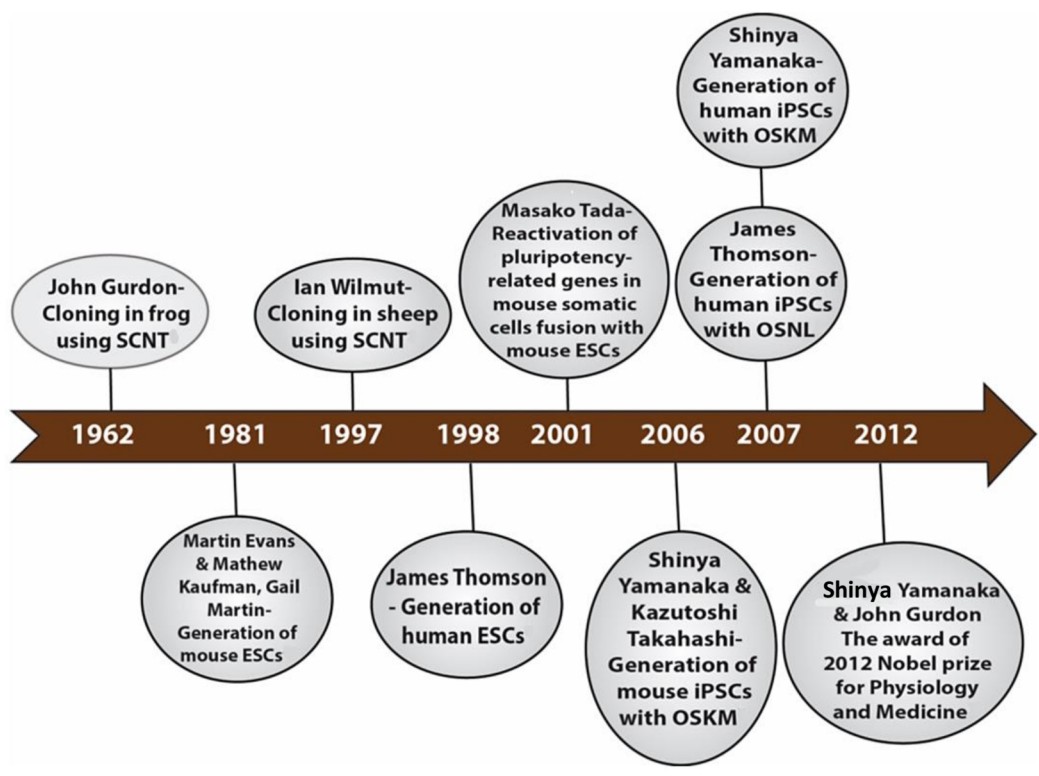

**Figure 1 Historical timeline showing events that led to the development of iPSCs.**

the "OSKM factors" (*Takahashi & Yamanaka, 2006*). A year later in 2007, Yamanaka and his team applied the same reprogramming method for adult human fibroblast to generate human iPSCs (hiPSCs) and James Thomson's group reported the generation of the same hiPSCs though using a different delivery system, the lentivirus and a different set of four factors: Oct 3/4, Sox2, Nanog, and Lin 28 (*Takahashi et al., 2007*; *Yu et al., 2007*). For their remarkable revolutionary discoveries, Shinya Yamanaka and John B. Gurdon were awarded the 2012 Nobel prize in Physiology or Medicine (*Gurdon & Yamanaka, 2012*). Like ESCs, the iPSCs have a self-renewal capability in culture and can differentiate into cell types from all three germ cell layers (ectoderm, mesoderm, and endoderm). The iPSC technology holds great promise for personalized cell-based therapy, human disease modeling, and drug development and screening. However, this technology is by no means free of its challenges. The reprogramming efficiency is low and tedious, and there is associated risk of chromosomal instability and tumorigenesis from insertional mutagenesis due to the viral vector delivery method (*Takahashi & Yamanaka, 2006*; *Takahashi et al., 2007*; *Yu et al., 2007*). These drawbacks will have a significant impact on the clinical application of iPSCs.

Much progress has since been made to improve the efficiency of reprogramming and to reduce the risk associated with the technology. Novel strategies already employed to improve reprogramming include the inhibition of barriers to reprogramming, use of non-integrative delivery methods, overexpression of enhancing genes and the use

of certain small molecules which enhanced reprogramming. Factors that influence the reprogramming process have been studied, namely, the choice of the somatic cell source, reprogramming transcription factors, delivery methods and culture conditions. Extensive research on the molecular mechanisms of reprogramming has significantly improved its efficiency.

In this review, we provide an overview of the progress made in iPSC technology in the last decade. First, we briefly define iPSCs by providing a summary of Yamanaka's key findings and the characterization of iPSCs and then summarize the current knowledge on the molecular mechanism of reprogramming, the limitations and the various strategies employed to address the drawbacks of this technology. We will then briefly discuss the potential application of iPSCs in the field of disease modeling, drug development, and regenerative medicine.

## METHODS

The data for this review were obtained from Medline on OvidSP, which includes PubMed, Embase by the US National Library of Medicine as well as a search through the University of Bristol Library services.

### Search strategy

A thorough search was carried out by signing into Ovid, Wolters, and Kluwer portal and "All Resources" was selected. Three separate keywords were used for the search. The first search with the keyword "induced pluripotent stem cells" yielded a total number of 5,975 publications. The second search with the keyword "cellular reprogramming" gave a total number of 3,002 publications. The third search with the keyword "transcription factors" gave a total number of 299,870 publications.

A combination of the search for "induced pluripotent stem cells" using the Boolean operator "AND" with "cellular reprogramming" and "transcription factor" yielded a total number of 200 publications. We next hand screened these 200 publications to see those that fit into the inclusion criteria for this study, and we arrived at a total of 114 publications.

Furthermore, other data were included in this review, and these were obtained from the University of Bristol Library services using the search phrase "induced pluripotent stem cells," "cellular reprogramming" and "transcription factors." The publications generated were hand screened to fit the inclusion criteria, and 61 publications were selected. Also included were relevant references from previously selected publications as well as many other recommended publications. A total of 228 articles were reviewed.

### Inclusion criteria

The publications selected were thoroughly analyzed to ensure they focused on the study objectives which are on the molecular mechanism of cellular reprogramming of somatic cells into iPSCs using transcription factors and other small molecules. We included studies that focused on the barriers and enhancers of cellular reprogramming and those that emphasized the various novel strategies for enhancing the kinetics and

efficiency of the process. Also considered were articles on the limitations and potential of iPSCs and the progress made to address such limitations. Publications that included the role of genomic editing technology in the generation of iPSCs were also considered.

## GENERATION OF iPSCs: A BRIEF OVERVIEW

Briefly, iPSCs can be defined as "embryonic stem cell-like" cells derived from the reprogramming of adult somatic cells by the introduction of specific pluripotent-associated genes. Prior to the discovery of iPSCs, ESCs which are derived from the inner cell mass (ICM) of a blastocyst of pre-implantation stage embryo, was the most well-known pluripotent stem cells. Just like ESCs, iPSCs can proliferate extensively in culture and can give rise to the three germ cell layers, namely, endoderm, mesoderm, and ectoderm.

Takahashi and Yamanaka set out to identify the genes that help in the maintenance of pluripotency in mouse ES cells. Their search led to a list of 24 candidate-reprogramming factors chosen for their links to ES-cell pluripotency. A screening method was developed to test a pool of 24 pluripotency-associated candidate factors for the ability to induce pluripotency. These genes were transduced into mouse embryonic fibroblasts (MEFs) using a retroviral delivery system. The mouse fibroblast was generated by the fusion of the mouse F-box only protein 15 (Fbxo15) gene locus with a β-galactosidase (β-geo) cassette. The expression of β-geo is used as a reporter of Fbxo15 expression and activity, as cells expressing β-geo are resistant to the selection marker geneticin (G418). The ESC-specific Fbxo-15 locus is not expressed in normal somatic cells which are not resistant to G418 treatment. The Fbxo15-β-geo MEFs were used to screen the pool of 24 transcription factors by transducing different combinations of the candidate genes and assessing the capability of the MEFs to survive in G418 treatment (Fig. 2). Consecutive rounds of elimination of each factor then led to the identification of a minimal core set of four genes, comprising Oct3/4, Sox2, Klf4, and c-Myc (OSKM cocktail/factors) (*Takahashi & Yamanaka, 2006*). These factors were already shown to be important in early embryonic development and vital for ES cell identity (*Avilion et al., 2003*; *Cartwright et al., 2005*; *Li et al., 2005*; *Niwa, Miyazaki & Smith, 2000*). The reprogrammed cell colonies, which were named as iPSCs, demonstrated ES cell-like morphology, express major ES cell marker genes like SSEA-1 and Nanog and formed teratomas upon injection into immunocompromised mice (*Takahashi & Yamanaka, 2006*) (Table 1).

Takahashi and Yamanaka demonstrated that ectopic expression of defined transcription factors was able to reprogram mouse fibroblasts back to a pluripotent state thus circumventing the ethical concerns surrounding the use of ESCs. However, these "first generation" iPSCs demonstrated a lower level of key ES pluripotency gene expression and failed to generate adult chimeras or contribute to the germline (*Takahashi & Yamanaka, 2006*). These latter characteristics suggest that the iPSCs were only partially reprogrammed. In 2007, Yamanaka and other laboratories modified the induction protocols to generate fully reprogrammed iPSCs that are competent for adult chimera and germline transmission (*Wernig et al., 2007*; *Okita, Ichisaka & Yamanaka, 2007*; *Maherali et al., 2007*). The technology has also been successfully translated to human fibroblasts

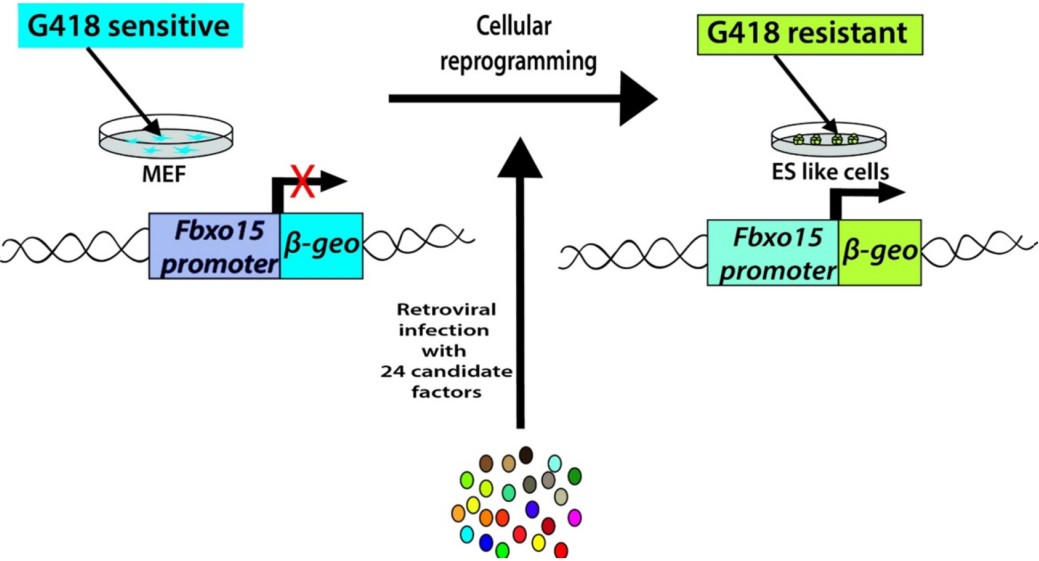

**Figure 2 Generation of iPSCs from MEF cultures via 24 factors by Yamanaka.**

(*Takahashi et al., 2007*; *Yu et al., 2007*; *Park et al., 2008a*) and then to other somatic cell types, such as pancreatic β cells (*Stadtfeld, Brennand & Hochedlinger, 2008*), neural stem cells (*Eminli et al., 2008*; *Kim et al., 2008*), stomach and liver cells (*Aoi et al., 2008*), mature B lymphocytes (*Hanna et al., 2008*), melanocytes (*Utikal et al., 2009a*), adipose stem cells (*Sun et al., 2009*) and keratinocytes (*Maherali et al., 2008*), demonstrating the universality of cellular reprogramming. The advantages of iPSC technology are its reproducibility and simplicity, thus encouraging many laboratories to modify and improve upon the reprogramming technique. Consequently, remarkable progress has been made in the last decade in the field of iPSC technology.

## TECHNICAL ADVANCES AND PROGRESS IN iPSC GENERATION

If iPSCs are to fulfill their promise (that they are viable and possibly superior substitutes for ESCs in disease modeling, drug discovery, and regenerative medicine), limitations and obstacles on the road to their clinical application need to be overcome. The initial reports of iPSC generation were inefficient (~0.001–1%) (*Takahashi & Yamanaka, 2006*; *Takahashi et al., 2007*; *Okita, Ichisaka & Yamanaka, 2007*; *Lowry et al., 2008*; *Huangfu et al., 2008b*), that is, on average only one out of 10,000 somatic cells formed iPSCs. The overexpression of oncogenes such as c-Myc and Klf4 during the generation of iPSCs raises safety concerns. Indeed, in the original report of germline-competent iPSCs, ~20% of the offspring developed tumors attributable to the reactivation of the c-Myc transgene (*Okita, Ichisaka & Yamanaka, 2007*). Furthermore, there is the risk of insertional mutagenesis due to virus-based delivery methods (*Takahashi & Yamanaka, 2006*; *Takahashi et al., 2007*; *Yu et al., 2007*). Much progress has been made in the past decade to address these limitations and to improve the reprogramming technique. New methods for

| Table 1 The characterization of iPSCs. | |
|---|---|
| Morphology | Flat, cobblestone-like cells, ES like morphology |
| | Tightly packed colonies with sharp edges |
| Pluripotency markers | Alkaline phosphatase assay (as a live marker) |
| | Increase levels of pluripotency proteins such as Oct4, Nanog, SSEA3/4, TRA-1-60, and TRA-1-81 |
| Differentiation potential | Teratoma formation—can form ectoderm, mesoderm, and endoderm, the three germ layers |
| | Embryoid body formation—can form ectoderm, mesoderm, and endoderm, the three germ layers |
| Genetic analyses | Diploid karyotype |
| | Transgene silencing after reprogramming |
| Epigenetic analyses | DNA methylation of lineage-committed genes |
| | DNA demethylation of key pluripotency genes like Oct4, Sox2, Nanog |

Note:
Adapted from *Brouwer, Zhou & Nadif Kasri (2016)*.

induced reprogramming have been developed. The following sections present an overview of the advancement made to improve the reprogramming technique, with emphasis on the reprogramming factors and the delivery systems for iPSC generation.

## Reprogramming factors

The conventional OSKM cocktail by Yamanaka's group has been used extensively by researchers on a wide range of human somatic cells and delivery systems (*Gonzalez, Boue & Izpisua Belmonte, 2011*). Thomson's group provided an alternative combination of four factors: Oct 3/4, Sox2, Nanog, and Lin 28 (OSNL) (*Yu et al., 2007*). The generation of iPSCs by Yamanaka's and Thomson's groups using different cocktails of transcription factors may suggest that different transcription factors activate the same reprogramming pathway by reinforcing each other's synthesis. The OSKM and OSNL reprogramming cocktails have proved efficient on a wide range of delivery systems, albeit at a variably low-efficiency rate (*Gonzalez, Boue & Izpisua Belmonte, 2011*; *Yakubov et al., 2010*). Consequently, researchers have sought to discover new molecules that will enhance the reprogramming technique and improve its efficiency (Table 2). We will refer to these molecules as reprogramming "enhancers." Some other molecules discovered are "barriers" of reprogramming technique. So the strategy employed to increase the efficiency of reprogramming includes the inhibition of such barriers and the overexpression and administration of the enhancers.

### Pluripotency-associated transcription factors

Many of the transcription factors used for reprogramming somatic cells are part of a core pluripotency circuitry. These factors are pluripotency-associated genes expressed early during embryonic development and are involved in the maintenance of pluripotency and self-renewal. The expression of other pluripotency-associated genes along with the minimal pluripotency factors (OSKM) can enhance the reprogramming efficiency or even

**Table 2 Reprogramming factors capable of reprogramming human cells.**

| Reprogramming factors | Function | Affected pathway | Effect on pluripotency | References |
|---|---|---|---|---|
| Oct4 | Maintenance of pluripotency and self-renewal | Core transcriptional circuitry | + | *Takahashi et al. (2007)* |
| Sox2 | Maintenance of pluripotency and self-renewal | Core transcriptional circuitry | + | *Takahashi et al. (2007)* |
| Klf4 | Maintenance of pluripotency and self-renewal | Core transcriptional circuitry | + | *Dang, Pevsner & Yang (2000)*, *Nakatake et al. (2006)* and *Guo et al. (2009)* |
| c-Myc | Maintenance of pluripotency and self-renewal | Core transcriptional circuitry | + | *Takahashi et al. (2007)* |
| Lin28 | Maintenance of pluripotency, translational enhancer, inhibits let7 | Core transcriptional circuitry | + | *Yu et al. (2007)* and *Buganim et al. (2012)* |
| Nanog | Maintenance of pluripotency and self-renewal | Core transcriptional circuitry | + | *Yu et al. (2007)* and *Buganim et al. (2012)* |
| Sall4 | Maintenance of pluripotency and self-renewal | Core transcriptional circuitry | + | *Tsubooka et al. (2009)* and *Buganim et al. (2012)* |
| Utf1 | Maintenance of pluripotency | Core transcriptional circuitry | + | *Zhao et al. (2008)* and *Buganim et al. (2012)* |
| p53 | Induces senescence, tumor suppressor | Apoptosis/cell cycle | − | *Kawamura et al. (2009)*, *Marion et al. (2009)*, *Utikal et al. (2009b)*, *Hong et al. (2009)* and *Banito et al. (2009)* |
| p21 | Induces senescence, tumor suppressor | Apoptosis/cell cycle | + | *Hong et al. (2009)*, *Banito et al. (2009)* and *Li et al. (2009a)* |
| MDM2 | p53 inhibitor | Apoptosis/cell cycle | + | *Hong et al. (2009)* |
| REM2 | p53 inhibitor | Apoptosis/cell cycle | + | *Edel et al. (2010)* |
| Cyclin D1 | Stimulates E2F/G1-S cell cycle transition | Apoptosis/cell cycle | + | *Edel et al. (2010)* |
| SV40 large T antigen | Inhibits p53 tumor suppression | Apoptosis/cell cycle | + | *Mali et al. (2008)* |
| DOT1L | Histone H3K79 methyltransferase | Chromatin remodeling | − | *Onder et al. (2012)* |
| MBD3 | Histone deacetylation, chromatin remodeling | Chromatin remodeling | − | *Rais et al. (2013)* |
| Sirt6 | Chromatin remodeling/ telomere maintenance | Chromatin remodeling | + | *Sharma et al. (2013)* |
| RCOR2 | Facilitates histone demethylation | Chromatin remodeling | + | *Yang et al. (2011)* |
| **Non-coding RNA** | | | | |
| miR367 | Inhibits EMT | TGFβ | + | *Anokye-Danso et al. (2011)* |
| LincRNA-ROR | Regulates expression of core transcriptional factors | Core transcriptional circuitry | + | *Loewer et al. (2010)*, *Wang et al. (2013)*, *Melton, Judson & Blelloch (2010)* and *Worringer et al. (2014)* |
| miR302 | Inhibits EMT/stimulates oct4 expression | TGFβ; Core transcriptional circuitry; apoptosis | + | *Anokye-Danso et al. (2011)*, *Lin et al. (2010, 2011)* and *Subramanyam et al. (2011)* |

| Reprogramming factors | Function | Affected pathway | Effect on pluripotency | References |
|---|---|---|---|---|
| miR766 | Inhibits Sirt6 | Chromatin remodeling | − | *Sharma et al. (2013)* |
| miR200c | Inhibits EMT/TGFβ pathway | TGFβ | + | *Miyoshi et al. (2011)* |
| miR369 | Inhibits EMT/TGFβ pathway | TGFβ | + | *Miyoshi et al. (2011)* |
| miR372 | Inhibits EMT/TGFβ pathway | TGFβ | + | *Subramanyam et al. (2011)* |
| Let7 | Regulates expression of core transcriptional factors and prodifferentiaion genes | Core transcriptional circuitry/TGFβ | − | *Loewer et al. (2010)*, *Wang et al. (2013)*, *Melton, Judson & Blelloch (2010)* and *Worringer et al. (2014)* |
| **Small molecules** | | | | |
| Vitamin C | Alleviates cell senescence/ antioxidant | Hypoxia response | + | *Wang et al. (2011)*, *Esteban et al. (2010)* and *Chung et al. (2010)* |
| Valproic acid | Inhibits histone deacetylases | Chromatin remodeling | + | *Huangfu et al. (2008a)* |
| CHIR99021 | GSK3-inhibitor | PI3k; Wnt/β-catenin | + | *Li et al. (2009b)* |
| Parnate | Lysine-specific demethylase 1 inhibitor | Chromatin remodeling | + | *Li et al. (2009b)* |
| BIX-01294 | Methyltransferase G9a inhibitor | Chromatin remodeling | + | *Feldman et al. (2006)* and *Shi et al. (2008)* |
| 5-Azacytidine | DNA methyltransferase inhibitor | Chromatin remodeling | + | *Huangfu et al. (2008a)* |
| Trichostatin A | Inhibits histone deacetylases | Chromatin remodeling | + | *Huangfu et al. (2008a)* |

**Note:**
Adapted from *Brouwer, Zhou & Nadif Kasri (2016)*.

replace some of the reprogramming factors. For example, the expression of undifferentiated embryonic cell transcription factor 1 (UTF1) or sal-like protein 4 (SALL4) with OSKM/OSK, improved the reprogramming efficiency (*Zhao et al., 2008*; *Tsubooka et al., 2009*). Non-coding RNA's like LincRoR and Let7 are involved in the regulation of expression of core transcriptional factors. LincRoR is a reprogramming enhancer while Let7 acts as a barrier by blocking the activation of pluripotency factors c-Myc, Lin 28, and SALL4. Thus, Let7 inhibition and the expression of LincRoR both enhance reprogramming efficiency (*Loewer et al., 2010*; *Wang et al., 2013*; *Melton, Judson & Blelloch, 2010*; *Worringer et al., 2014*). Nanog and Lin 28 can replace Klf4 and c-Myc respectively, and estrogen-related receptor beta (ESRRβ) can replace Klf4 (*Yu et al., 2007*; *Feng et al., 2009*). A recent single-cell gene expression study of partially reprogrammed cells showed that SALL4, ESRRβ, Nanog and Lin 28 (rather than OSKM) was enough for reprogramming fibroblasts into iPSCs, albeit with low efficiency (*Buganim et al., 2012*). These observations suggest that most of these enhancer genes are possibly part of the reprogramming circuitry network activated by OSKM. Consequently, a detailed analysis of the downstream targets of OSKM may help us understand the molecular mechanisms of reprogramming, thus opening the way to increasing its efficiency.

### Cell cycle-regulating genes

As they move toward pluripotency, somatic cells also gain the ability to proliferate indefinitely. Not surprisingly, two of the minimal pluripotency factors, c-Myc, and Klf4,

are oncogenes that enhance cellular proliferation. Apparently, there will be other regulators in this cell cycle pathway. The p53 tumor suppressor protein promotes senescence and inhibits growth, thus having an inhibitory effect on iPSCs generation (*Kawamura et al., 2009*; *Marion et al., 2009*; *Utikal et al., 2009b*; *Hong et al., 2009*; *Banito et al., 2009*). Many studies have shown that p53 inhibition can greatly enhance reprogramming efficiency (*Kawamura et al., 2009*; *Marion et al., 2009*; *Utikal et al., 2009b*; *Hong et al., 2009*; *Banito et al., 2009*). Cell cycle-dependent kinase inhibitors like INK4A and ARF (which are linked to the p53–p21 pathway) can block iPSC reprogramming (*Li et al., 2009a*). Conversely, overexpression of p53 inhibitor proteins (such as SV40 large T antigen, REM2, and MDM2), increased the efficiency of reprogramming (up to 23-fold compared to OSKM alone) (*Park et al., 2008a*; *Hong et al., 2009*; *Mali et al., 2008*; *Edel et al., 2010*). So researchers have used the strategy of overexpressing reprogramming enhancers to eliminate the barriers on the road toward pluripotency.

### Epigenetic modifiers

The reprogramming of somatic cells into iPSCs is characterized by epigenetic changes, from DNA methylation to histone modifications. Chromatin remodeling is a rate-limiting step in the reprogramming process, and thus researchers have studied chemical compounds that modify the epigenetic process (*Huangfu et al., 2008a*). For example, DNA methyltransferase inhibitor 5-azacytidine and histone deacetylase (HDAC) inhibitors (like suberoylanilide hydroxamic acid (SAHA), trichostatin A (TSA) and valproic acid (VPA)) enhanced reprogramming efficiency in MEFs (*Huangfu et al., 2008a*). VPA also promotes somatic cell reprogramming with Oct4 and Sox2 alone (*Huangfu et al., 2008b*). The combination of CHIR99021 (a GSK3 inhibitor) with Parnate (a lysine-specific demethylase one inhibitor) causes the reprogramming of human keratinocytes with only Oct4 and Klf4 (*Li et al., 2009b*). Similarly, G9a histone methyltransferase promotes epigenetic repression of Oct4 during embryonic development (*Feldman et al., 2006*), which is why a G9a inhibitor (BIX-01294) enhances MEF reprogramming with only Oct4 and Klf4 (*Shi et al., 2008*). Disruptor of telomeric silencing 1-like (DOT1L) (*Onder et al., 2012*), methyl-CpG binding domain protein 3 (MBD3) (*Rais et al., 2013*), rest corepressor 1 (RCOR2) (*Yang et al., 2011*), sirtuin 6 (Sirt6), and miR766 (a Sirt6 inhibitor) (*Sharma et al., 2013*) are all involved in chromatin remodeling, thus affecting the efficiency of reprogramming when inhibited or overexpressed. Vitamin C improves cellular reprogramming efficiency, in part by promoting the activity of histone demethylases JHDM1A (KDM2A) and JHDM1B (KDM2B) (*Wang et al., 2011*), alleviating cell senescence (*Esteban et al., 2010*) and inducing DNA demethylation (*Chung et al., 2010*).

In conclusion, microRNA (miRNA) have been used to increase reprogramming efficiency. The miRNA's mostly work by inhibiting the TGFβ signaling pathway, thereby inhibiting the epithelial to mesenchymal transition (EMT). The combination of miR-291-3p, miR-294, and miR-295 with OSK cocktail promotes iPSC generation (*Judson et al., 2009*). More recently, miR302, miR367, miR369, miR372, and miR200c have been used either alone or in combinations to enhance the reprogramming process in

humans by replacing the traditional OSKM nuclear factors (*Anokye-Danso et al., 2011*; *Lin et al., 2010*, *2011*; *Miyoshi et al., 2011*; *Subramanyam et al., 2011*). The miRNAs can specifically target multiple pathways thus reducing the need and amount of transcription factors for reprogramming (*Subramanyam et al., 2011*). In the near future, miRNA-based reprogramming may provide a more effective way of cellular reprogramming than traditional nuclear factor (OSKM) methods.

## Delivery methods

A number of different delivery methods have been used to introduce reprogramming factors into somatic cells (Fig. 3). The reprogramming methods can be grouped into two categories—**integrative systems** (involving the integration of exogenous genetic material into the host genome) and **non-integrative systems** (involving no integration of genetic material into the host genome). The integrative delivery methods include the use of viral vectors (retrovirus, lentivirus, and inducible lentivirus) and non-viral vectors (linear/plasmid DNA fragments and transposons). Similarly, the non-integrative delivery methods include the use of viral vectors (adenovirus and Sendai virus) and non-viral vectors (episomal DNA vectors, mRNA, and proteins). This section is focused on the reprogramming methods currently available.

### Integrative delivery systems

#### Viral integrative vectors

**Retroviruses** were used for the delivery of transcription factors in the original studies on iPSC generation (*Takahashi & Yamanaka, 2006*; *Takahashi et al., 2007*; *Wernig et al., 2007*; *Okita, Ichisaka & Yamanaka, 2007*; *Maherali et al., 2007*). Retroviruses are an efficient and relatively easy form of the delivery system. They require an actively dividing somatic cell to integrate well in the genome. iPSC is considered to be fully reprogrammed only after the upregulation of endogenous pluripotency genes and the downregulation or silencing of the integrated transgene expression. Though retroviral vectors are usually silenced in ESCs (*Jahner et al., 1982*; *Stewart et al., 1982*) and iPSCs (*Park et al., 2008a*; *Nakagawa et al., 2008*), the silencing is not always efficient, and the silenced transgenes may be reactivated later on. They can integrate randomly into the host genome leading to an increased risk of insertional mutagenesis. Certainly, in the original report of germline-competent iPSCs, ∼20% of the offspring developed tumor attributable to the reactivation of c-Myc transgene (*Okita, Ichisaka & Yamanaka, 2007*).

**Lentivirus** has also been successfully used for the introduction of transgenes during cellular reprogramming (*Yu et al., 2007*; *Blelloch et al., 2007*). Like retroviral vectors, lentivirus integrates into the host genome with the risk of insertional mutagenesis, and inefficient silencing and transgene reactivation are possible. Unlike retroviruses, they can integrate into both dividing and non-dividing cells. Thus iPSCs can be generated from most somatic cell types. The original studies on iPSC generation by Yamanaka involved the use of different types of retroviruses, each delivering only one type of transcription factor (*Takahashi et al., 2007*). This can create many uncontrollable integration events with increased risks of transgene reactivation, inefficient transgene silencing and

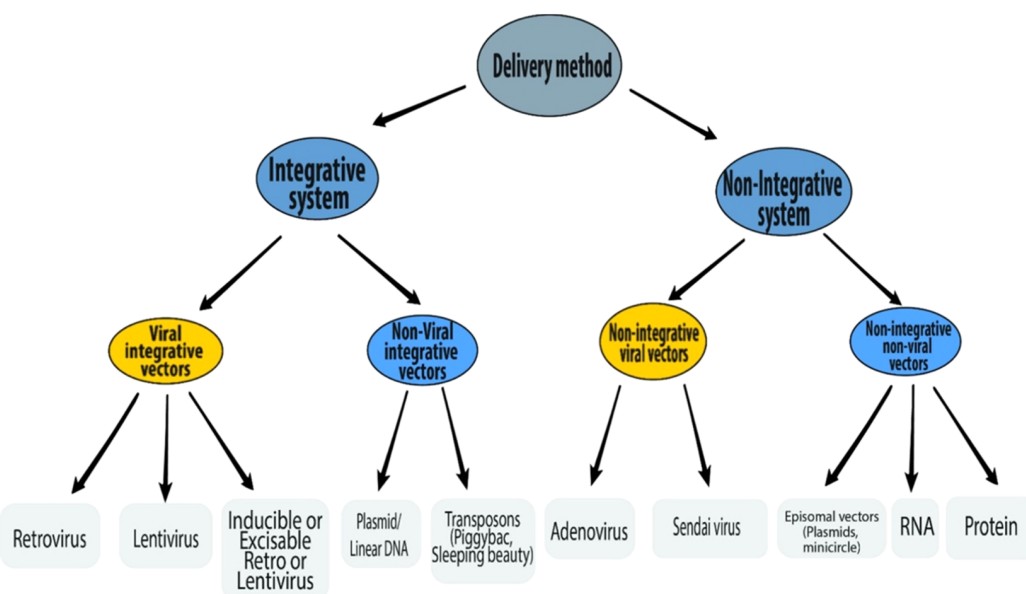

**Figure 3 Schematic representation of various delivery methods of iPSC induction.**

diminished efficiency of reprogramming. The creation of ***polycistronic*** viral vectors (for retrovirus (*Rodriguez-Piza et al., 2010*) and lentivirus (*Carey et al., 2009*; *Sommer et al., 2009*)) allowed the expression of all reprogramming factors driven by a single promoter, with the genes separated by self-cleaving peptide sequences. This method remarkably reduces the number of genomic insertions thus improving the safety and efficiency of the reprogramming process. Moreover, the introduction of both the excisable (***Cre/loxP***) vector system (*Soldner et al., 2009*; *Chang et al., 2009*) and inducible (***tetracycline/ doxycycline-inducible***) vector system (*Maherali et al., 2008*; *Hockemeyer et al., 2008*; *Wernig et al., 2008a*; *Staerk et al., 2010*) has allowed for a better control of transgene expression thus reducing the effects of inefficient silencing and transgene reactivation.

*Non-viral integrative vectors*
An alternative to viral vectors is the transfection of **DNA (plasmid/linear)** into cells using liposomes or electroporation. Using this method, the transduction efficiency is much lower with only a few cells capturing the full set of reprogramming factors. However, the use of polycistronic vectors to express all cDNAs from a single promoter has helped to improve the reprogramming efficiency. *Kaji et al. (2009)* successfully generated iPSCs from mouse fibroblasts with a non-viral polycistronic vector combined with an excisable Cre/loxP system for deleting the reprogramming construct.

**Transposons**. *Kaji et al. (2009)* and *Woltjen et al. (2009)* applied the non-viral single vector system for the generation of human iPSCs using a **piggybac** (PB) transposon-based delivery system. The PB is a mobile genetic element which includes an enzyme PB transposase (that mediates gene transfer by insertion and excision). Co-transfection of a donor plasmid (transposon) with a helper plasmid expressing the transposase enzyme leads to the efficient integration of the transposon (*Gonzalez, Boue & Izpisua*

*Belmonte, 2011*). Once the reprogramming is achieved, the enzyme can precisely delete the transgenes without any genetic damage thus avoiding the risk of insertional mutagenesis. Drawbacks to the use of PB systems include the risks of integrating back into the genome, and the potential that the human genome contains endogenous PB transposon elements which may be acted upon by the transposase enzyme essential for the transgene excision (*Newman et al., 2008*; *Feschotte, 2006*; *Grabundzija et al., 2010*; *Brouwer, Zhou & Nadif Kasri, 2016*). The recent introduction of another transposon, **sleeping beauty** (SB), has helped to overcome many of the limitations of the PB transposon (*Grabundzija et al., 2013*; *Davis et al., 2013*). SB integrates less compared to the PB, and there are no SB-like elements in the human genome. However, the reprogramming efficiency of transposons is low compared to viral vectors, and their use involves multiple rounds of excision, thus increasing the risk of re-integration.

Overall, integrative delivery systems come with a risk of integration into the genome leading to insertional mutagenesis. This lack of safety may limit their therapeutic application. Non-integrative delivery systems will later address this major limitation.

### Non-integrative delivery systems

*Non-integrative viral vectors*
*Stadtfeld et al. (2008a)* reported the generation of the first integration-free iPSCs from adult mouse hepatocytes using nonintegrating **adenovirus**. Transgene-free iPSCs were later generated from human fibroblasts by *Zhou & Freed (2009)* using similar adenoviral vectors. However, the reprogramming process requires multiple viral infections, and the production of adenovirus is very labor-intensive. Most importantly, the reprogramming efficiency using adenoviruses is several orders of magnitude lower compared to lenti- or retroviruses.

Another non-integrating viral vector that has been successfully used for iPSC generation is the **Sendai virus** (SeV) (*Fusaki et al., 2009*; *Seki et al., 2010*; *Ban et al., 2011*; *Nishishita et al., 2012*; *Ono et al., 2012*; *Seki, Yuasa & Fukuda, 2012*; *Macarthur et al., 2012*). These are very efficient in transferring genes (in the form of negative-strand single-stranded RNA) into a wide range of somatic cells (*Li et al., 2000*; *Tokusumi et al., 2002*; *Inoue et al., 2003*; *Nakanishi & Otsu, 2012*). Although they are very effective, the viral vector's RNA replicase is very sensitive to the transgene sequence content. Additionally, because they constitutively replicate, these vectors may be difficult to eliminate from the somatic cells (*Fusaki et al., 2009*). A new improved Sendai virus (**SeV dp**) has since been developed (*Nishimura et al., 2011*; *Kawagoe et al., 2013*).

*Non-integrative non-viral delivery*
**Episomal vectors** provide another alternative to the integrative–defective viruses. Episomes are extrachromosomal DNAs capable of replicating within a cell independently of the chromosomal DNA. The reprogramming factors can be directly and transiently transfected into the somatic cells using episomal vectors as **plasmids** (*Okita et al., 2008*, *2010*, *2011*; *Yu et al., 2009*; *Gonzalez et al., 2009*; *Cheng et al., 2012*; *Montserrat et al., 2011*;

Si-Tayeb et al., 2010) or as **minicircle** DNA (*Jia et al., 2010*; *Narsinh et al., 2011*). Unlike retro- and lentiviruses, this technique is relatively simple and easy to use and does not involve integration into the host genome. However, since episomal vector expression is only transient, they require multiple transfections. In general, their reprogramming efficiency is low, although when compared to its plasmids, the minicircle DNA has a higher transfection efficiency (probably due to it is smaller size) and a longer ectopic expression of the transgenes (due to lowered silencing mechanisms) (*Chen et al., 2003*; *Chen, He & Kay, 2005*).

**RNA delivery.** iPSCs have been generated by the direct delivery of synthetic mRNA into somatic cells (*Warren et al., 2010*, *2012*). This method has the highest reprogramming efficiency when compared with other non-integrative delivery systems. RNA has short half-lives. Thus repeated transfection is required to sustain the reprogramming process. RNA-based methods are also highly immunogenic (*Brouwer, Zhou & Nadif Kasri, 2016*).

**Protein delivery.** Reprogramming factors can be directly delivered as recombinant proteins into somatic cells for iPSC generation (*Kim et al., 2009*; *Zhou et al., 2009*). The reprogramming efficiency is low and repeated transfection is also required to maintain the intracellular protein level for reprogramming.

Overall, integrative delivery methods have a higher reprogramming efficiency than non-integrating methods, but they are less safe due to the risk of insertional mutagenesis. Therefore, the use of non-integrating methods will appeal more for iPSC generation and use in a clinical setting.

## MOLECULAR MECHANISM OF INDUCED PLURIPOTENCY

The reprogramming of somatic cells into iPSCs is a long and complex process involving the activation of ES-cell-specific transcription network, combinatorial overexpression of multiple transcription factors and epigenetic modifications. Understanding the molecular mechanisms of cellular reprogramming is critical for the generation of safe and high-quality iPSCs for therapeutic applications. This section reviews the molecular mechanisms leading to induced pluripotency.

### The fantastic four (OSKM)

Takahashi and Yamanaka showed that four exogenous reprogramming factors, Oct4, Sox2, Klf4, and c-Myc, all have key roles in iPSC generation (*Takahashi & Yamanaka, 2006*). They discovered Oct4, Sox2, Klf4, and c-Myc were essential for iPSC generation while Nanog was dispensable (*Takahashi & Yamanaka, 2006*). Though exogenous Nanog (not part of the "fantastic four") is not an essential factor and is not required to initiate the reprogramming process, it is possible that exogenous Oct 4, Sox2, and other reprogramming factors induce expression of endogenous Nanog to levels adequate to achieve full reprogramming (*Jaenisch & Young, 2008*; *Scheper & Copray, 2009*).

Genetic studies have shown that Oct4, Sox2, and Nanog (OSN) are key regulators of embryonic development and they are critical for pluripotency maintenance (*Masui et al., 2007*; *Chambers et al., 2003*, *2007*; *Avilion et al., 2003*; *Nichols et al., 1998*;

*Mitsui et al., 2003*). These factors are expressed both in pluripotent ESCs and in the ICM of blastocysts. Oct 3/4, Sox2, and Nanog knockout embryos die at the blastocyst stage and when cultured in vitro, their ESCs lose pluripotency and differentiate (*Avilion et al., 2003*; *Nichols et al., 1998*; *Mitsui et al., 2003*; *Chambers et al., 2007*). Klf4 plays key roles in cellular processes, like development, proliferation, differentiation, and apoptosis (*Dang, Pevsner & Yang, 2000*). It is expressed in ESCs and can interact with Oct4–Sox2 complexes to activate certain ESCs genes (*Nakatake et al., 2006*). Klf4 can revert epiblast-derived stem cells to the ESC state (*Guo et al., 2009*). Its interaction with Oct4–Sox2 complexes and its tumor suppressor activity are thought to be important in iPSCs generation. c-Myc is a potent oncogene associated with apoptosis, cell proliferation, and cell cycle regulation (*Dang et al., 2006*; *Lebofsky & Walter, 2007*; *Patel et al., 2004*). Though iPSCs can be generated without Klf4 and c-Myc, the marked reduction in the efficiency of the process greatly emphasizes their importance in cellular reprogramming.

## Autoregulatory loops driving pluripotency

Experimental studies using chromatin immunoprecipitation and genome-wide localization analysis in human and murine ESCs to identify genes occupied by Oct4, Sox2, and Nanog have provided a better understanding of how these transcription factors contribute to pluripotency (*Boyer et al., 2005*; *Loh et al., 2006*). The studies reveal that Oct4, Sox2, and Nanog bind together to activate the promoters of both their genes and those of each other, hence forming an autoregulatory loop. The three factors function cooperatively to maintain their expression, thus enhancing the stability of pluripotency gene expression. Since the initial hypothesis, several other studies have provided strong verifiable evidence for the existence of the autoregulatory circuitry (*Masui et al., 2007*; *Chew et al., 2005*; *Kuroda et al., 2005*; *Okumura-Nakanishi et al., 2005*; *Rodda et al., 2005*).

## Transcriptional regulatory network

The experimental studies also demonstrated that Oct4, Sox2, and Nanog target several hundred other ESC genes, collectively co-occupying these genes cooperatively to maintain the transcriptional regulatory network required for pluripotency (*Boyer et al., 2005*; *Loh et al., 2006*). This may explain why efficient iPSC generation seems to require the combinatorial overexpression of multiple transcription factors. The cascades of genes targeted were found to be both transcriptionally active and inactive genes (Table 3). The actively transcribed genes all have a key role in the maintenance of ESC pluripotency and self-renewal. They include various ESC transcription factors, chromatin modifying enzymes, and ESC-signal transduction genes. Conversely, the inactive genes are essentially developmental transcription factors that are silent in ESCs, whose expression is associated with cellular differentiation and lineage commitment (*Boyer et al., 2005*; *Loh et al., 2006*). Altogether, Oct4, Sox2, and Nanog appear to be master regulators of induced pluripotency by enhancing transcription of pluripotency genes, while at the same time silencing genes related to development and differentiation. Therefore, to achieve

**Table 3** The Oct4, Sox2 and Nanog trio contributes to ES cell pluripotency by repressing genes linked to lineage commitment and activating genes involved in pluripotency.

| Transcriptionally active genes | | Transcriptionally inactive genes | |
| --- | --- | --- | --- |
| Genes | Role of activated genes | Genes | Role of inactivated genes |
| Oct4, Sox2, Nanog | Key pluripotency genes | Pax6, Meis1, Hoxb1, Lhx5, Otx1, Neurog1 | Ectoderm development |
| Stat3, Hesx1, Zic3, Esrrb | ES cell transcriptions factors | Hand1, Dlx5, Myf5, Onecut1 | Mesoderm development |
| Tcf3, Fgf2, Lefty2, Skil | ES cell signaling | Isl1, Atbf1 | Endoderm development |
| Smarcad1, Myst3, Setdb1, Jarid2 | Epigenetic regulators | Esx1l | Extra-embryonic development |
| Rest | Inhibitor of neurogenesis | | |
| Rif1 | Telomere-associated protein | | |

pluripotency, the autoregulatory loops and the transcriptional regulatory network need to be resuscitated in reprogrammed somatic cells.

## Epigenetic changes during iPSC reprogramming

Induced pluripotent stem cells have a unique epigenetic signature that distinguishes them from differentiated somatic cells. Pluripotent stem cells have open, active chromatin conformations, with activating histone H3 lysine-4 trimethylation marks (H3K4me3), histone acetylation and hypomethylated DNA around the pluripotency genes. In contrast, lineage-commitment leads to the silencing of these pluripotency genes, with repressive H3K27me3 and H3K9me3 histone marks, hypermethylated DNA and a closed heterochromatin conformation. During the reprogramming process, the epigenetic signature of the somatic cell must be erased to adopt a stem cell-like epigenome. These epigenetic changes include chromatin remodeling, DNA demethylation of promoter regions of pluripotency genes, reactivation of the somatically silenced X chromosome and histone post-translational modifications (*Takahashi et al., 2007*; *Wernig et al., 2007*; *Maherali et al., 2007*; *Fussner et al., 2011*; *Buganim, Faddah & Jaenisch, 2013*; *Gonzalez & Huangfu, 2016*).

### DNA methylation in iPSC reprogramming

DNA methylation is an epigenetic barrier of iPSC generation (*Nishino et al., 2011*; *Doege et al., 2012*; *Gao et al., 2013*). The methylation occurs at the C5 position of cytosine on the target gene promoters in mammalian somatic cells (*Gonzalez & Huangfu, 2016*). Promoter DNA methylation is inversely associated with gene expression (*Bird, 2002*). The epigenome of iPSCs are transcriptionally active and are characterized by demethylation at the promoter regions of key pluripotency genes, like Oct4, Sox2, and Nanog. These genes are silenced by de novo DNA methylation during lineage commitment and differentiation. The methylation is established by de novo methyltransferases Dnmt3a and Dmnt3b and preserved by the maintenance methyltransferase Dnmt1 (*Smith & Meissner, 2013*). During reprogramming, the methylation marks are removed from these endogenous pluripotency genes to allow for their transcription, and tissue-specific genes are hypermethylated (*Gladych et al., 2015*; *Berdasco & Estellar, 2011*). Indeed, manipulation of the DNA and chromatin

modifications by certain small molecules can significantly improve iPSC formation (*Huangfu et al., 2008a*, *2008b*; *Li et al., 2009b*; *Feldman et al., 2006*) (see reprogramming factors–epigenetic modifiers). Likewise, mice lacking DNA methyltransferases remain non-viable or die within weeks (*Li, Bestor & Jaenisch, 1992*; *Okano et al., 1999*). These observations show that epigenetic modifications are key to cellular differentiation, and it is reasonable to conclude that these same events have to be reversed during induced reprogramming.

### Histone modifications in iPSC reprogramming

Histone modification patterns differ between pluripotent stem cells and differentiated somatic cells. The silencing of developmental genes in pluripotent stem cells is controlled remarkably. The differentiation-related genes carry "bivalent" domains (i.e., repressive histone H3 lysine-27 trimethylation marks (H3K27me3) and activating histone H3 lysine-4 trimethylation marks (H3K4me3)) in their genome loci (*Bernstein et al., 2006*). The H3K4me3 marks of the bivalent domains allow for transcription initiation of the developmental genes. Transcription of these genes is repressed by the action of Polycomb group, a family of proteins that regulate developmental gene expression through gene silencing by binding to repressive H3K27me3 marks. Thus, lineage-commitment genes with bivalent domains can have their expression quickly turned on or switched off via erasure of H3K27me3 or H3K4me3, respectively. The bivalent domains are almost exclusively found in pluripotent stem cells, and their restoration represents a vital step in the reprogramming process. During reprogramming, repressive H3K9me3 marks present on the endogenous pluripotency genes (Oct4, Sox2, and Nanog) are gradually replaced by the transcriptionally active H3K4me3 (*Gladych et al., 2015*). The loss of the H3K9me3 marks allows access of OSKM transgenes to their target regions thus activating the autoregulatory loop.

## Role of microRNAs in iPSC reprogramming

microRNA are small non-coding RNA molecules that bind to protein-coding messenger RNA (mRNA) to regulate their degradation or translation. They regulate gene expression by post-transcriptional gene silencing (*Bartel, 2004*). Some miRNA promote iPSC reprogramming (see reprogramming factors–epigenetic modifiers), while others are barriers to iPSC reprogramming. Let-7 miRNAs are expressed in somatic cells and upregulated in ES cell differentiation (*Roush & Slack, 2008*). Lin 28 (one of the factors used by *Thomson et al. (1998)* to substitute for c-Myc and Klf4) (*Yu et al., 2007*), promotes reprogramming by inhibiting let-7 miRNAs (*Viswanathan, Daley & Gregory, 2008*).

## The role of reprogramming factors in iPSC reprogramming

Following the introduction of exogenous OSKM factors into the somatic cells, exogenous Oct4 and Sox2 may directly induce the expression of endogenous Oct4, Sox2, and Nanog via the autoregulatory circuitry, through which they continue to maintain their expression. After that, these factors activate the pluripotent transcriptional network. Hence, the autoregulatory loop and the transcriptional network that are repressed in somatic cells, are now "resuscitated" during the reprogramming process.

c-Myc is a vital component of active chromatin and associates with histone acetyltransferase (HAT) complexes. Thus, it facilitates an open chromatin conformation through global histone acetylation, thereby allowing Oct4 and Sox2 to target their genomic loci (*Kim et al., 2008*; *Scheper & Copray, 2009*; *Knoepfler, Zhang & Cheng, 2006*). As a well-known oncogene, c-Myc facilitates the cancer-like transformation of somatic cells, conferring immortality and rapid proliferative potential on the pluripotent stem cells (*Yamanaka, 2007*). Thus, cellular division driven by c-Myc may provide somatic cells an opportunity to reset their epigenome, thereby enhancing their reprogramming (*Jaenisch & Young, 2008*). As was mentioned in reprogramming factors–cell cycle regulating genes, p53 tumor suppressor proteins have inhibitory effects on iPSC generation by promoting senescence, apoptosis and cell cycle inhibition (*Kawamura et al., 2009*; *Marion et al., 2009*; *Utikal et al., 2009b*; *Hong et al., 2009*; *Banito et al., 2009*). Hyperexpression of c-Myc can lead to increases in p53 levels, and Klf4 can block the resulting apoptotic effect of c-Myc by suppressing p53 levels (*Rowland, Bernards & Peeper, 2005*). Furthermore, Klf4 can suppress proliferation by activating p21 (a cyclin-dependent kinase inhibitor), and c-Myc can inhibit this anti-proliferative effect of Klf4 by suppressing p21 (*Zhang et al., 2000*; *Seoane, Le & Massague, 2002*). Thus, we can conclude that c-Myc and Klf4 are mutually complementary and a balance between their expression is necessary for successful reprogramming (*Scheper & Copray, 2009*; *Yamanaka, 2007*). The overall summary of the roles of reprogramming factors is shown in Fig. 4.

## Two-phase model of induced reprogramming: a gradual, stochastic process

Several studies have shown exactly how the ectopic expression of OSKM in somatic cells induces the transition to a pluripotent state (*Yamanaka, 2007*; *Brambrink et al., 2008*; *Stadtfeld et al., 2008b*; *Polo et al., 2012*; *Hansson et al., 2012*; *Buganim et al., 2012*). Based on these studies, we now know the order of events in the reprogramming process, and we can posit that the reprogramming process consists of two broad phases: An initial, stochastic **early** phase (phase 1) and a more deterministic and hierarchical **late** phase (phase 2) (Table 4).

### Phase 1

The earliest event in phase 1 is the **downregulation of lineage-specific genes**. This may be due to the direct repression effect of OSKM on these developmental genes or indirectly through the restoration of bivalent histone marks on the same genes (*Scheper & Copray, 2009*). The next event is the **upregulation of a subset of ESC-specific genes, such as alkaline phosphatase (AP), Fbx15, and SSEA1**. These two events may produce a partially reprogrammed iPSC with ESC-like morphology, but can quickly revert to the differentiated state once the transgene expression is terminated. The next step is the **global chromatin remodeling of the full array of pluripotency genes**. This event involves the gradual unfolding of condensed heterochromatin to form an open euchromatin conformation and the removal of repressive H3K9me3 histone marks. The latter event is brought on by the effect of c-Myc, Klf4, histones modification enzymes (acetyltransferases and demethylases) and other small molecules. The removal of the repressive histone

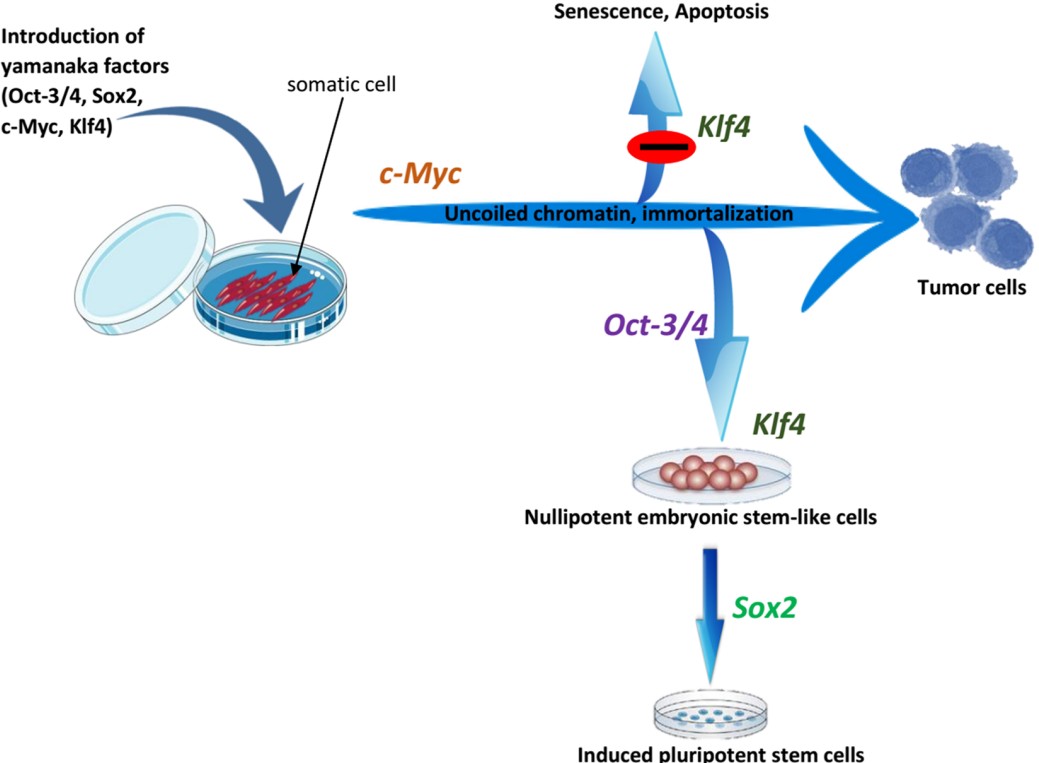

**Figure 4 The roles of OSKM factors in the induction of iPSCs.** Pluripotent stem cells are immortal with open and active chromatin structure. It is probable that c-Myc induce these two properties by binding to several sites on the genome and by the recruitment of multiple histone acetylase complexes. However, c-Myc also induces apoptosis and senescence and this effect may be antagonized by Klf4. Oct3/4 probably changes the cell fate from tumor cells to ES-like cells while Sox2 helps to drive pluripotency. Adapted from *Yamanaka (2007)*.                

marks requires multiple rounds of cell division and explains why reactivation of endogenous Oct4, Sox2, and Nanog occurs late in the reprogramming process.

### Phase 2

After the completion of global chromatin remodeling, exogenous Oct4 and Sox2 are now able to target and activate the loci of endogenous Oct4, Sox2, and Nanog genes leading to the **resuscitation of the autoregulatory loop**. The completion of chromatin remodeling at other pluripotency genes further leads to the gradual **restoration of the full ESC transcription network**. This leads to the establishment of full-blown pluripotency, characterized by reactivation of telomerase, inactivated X chromosome and ESC signaling cascades. As the autoregulatory loops continue to self-maintain the expression of the endogenous pluripotency genes, the **transgene silencing** previously initiated in phase 1 comes to completion. The pluripotent state is now completely dependent on the endogenous autoregulatory circuitry.

### iPSC reprogramming—an inefficient process

As mentioned above in technical advances and progress in iPSC generation, low reprogramming efficiency is one of the limitations of induced reprogramming (*Takahashi &*

**Table 4 Two-phase model of induced reprogramming.**

| Order of events | Phase 1 | Phase 2 |
|---|---|---|
| Step 1 | Downregulation of lineage genes by direct repression and restoration of bivalent domains | Resuscitation of autoregulatory loop |
| Step 2 | Activation of specific ES cell genes such as AP, Fbx15, and SSEA1 | Full reactivation of ES cell transcriptional network by reactivation of telomerase and ES cell signal cascades |
| Step 3 | Chromatin remodeling at pluripotency genes by the unfolding of condensed chromatin and the removal of repressive chromatin marks | Completion of transgene silencing |

*Yamanaka, 2006*; *Takahashi et al., 2007*; *Okita, Ichisaka & Yamanaka, 2007*; *Lowry et al., 2008*; *Huangfu et al., 2008b*). The *elite*, *stochastic* and *deterministic* models have been posited to explain the reason why only a small part of the transduced cells become pluripotent.

*Elite model.* This model postulates that only a few, rare, "elite" somatic cells (with stem cells characteristics) present within the somatic cell population, can be induced towards pluripotency (*Yamanaka, 2009*; *Takahashi & Yamanaka, 2016*). In contrast to these "special" cells, differentiated cells within the population are resistant to OSKM-mediated induction (Fig. 5A). Although somatic cell populations are heterogeneous and contain stem cells (*Goodell, Nguyen & Shroyer, 2015*), we now know that fully differentiated cells can be reprogrammed, thus disproving the elite model (*Stadtfeld, Brennand & Hochedlinger, 2008*; *Aoi et al., 2008*; *Hanna et al., 2008*). Most of the somatic cells initiate the reprogramming process, but the majority never complete it.

*Stochastic and deterministic models.* Assuming all somatic cells are transduced by the OSKM, the next path to pluripotency could occur by two mechanisms: a "**stochastic**" manner in which iPSCs appear at different, random, unpredictable periods; or a "deterministic" manner in which iPSCs appear at a fixed, predictable period (Figs. 5B and 5C). Both types of mechanism might be involved in the reprogramming process.

The generation of iPSCs requires a precise, limited-level expression of the transduced factors and the process involve tightly regulated levels of pluripotency genes. The specific stoichiometric balance of the OSKM factors is fundamental to successful reprogramming (*Tiemann et al., 2011*; *Yamaguchi et al., 2011*). Thus, maintaining this delicate balance appropriately can be a difficult, even rare event. Additionally, somatic cells have to overcome many barriers on the road to pluripotency (see two-phase model of induced reprogramming: a gradual, stochastic process). Furthermore, random transgene integration can result in heterogeneous transgene expression that is achieved by very few cells. The lower chance of completing these stochastic reprogramming events and the need to overcome reprogramming barriers altogether contribute to the low efficiency of reprogramming.

There are other variables that can affect the efficiency of induced reprogramming such as the choice of; reprogramming factors, delivery methods, donor cell types and culture conditions (*Gonzalez, Boue & Izpisua Belmonte, 2011*; *Brouwer, Zhou & Nadif Kasri, 2016*). We have already considered the effects of reprogramming factors and delivery methods

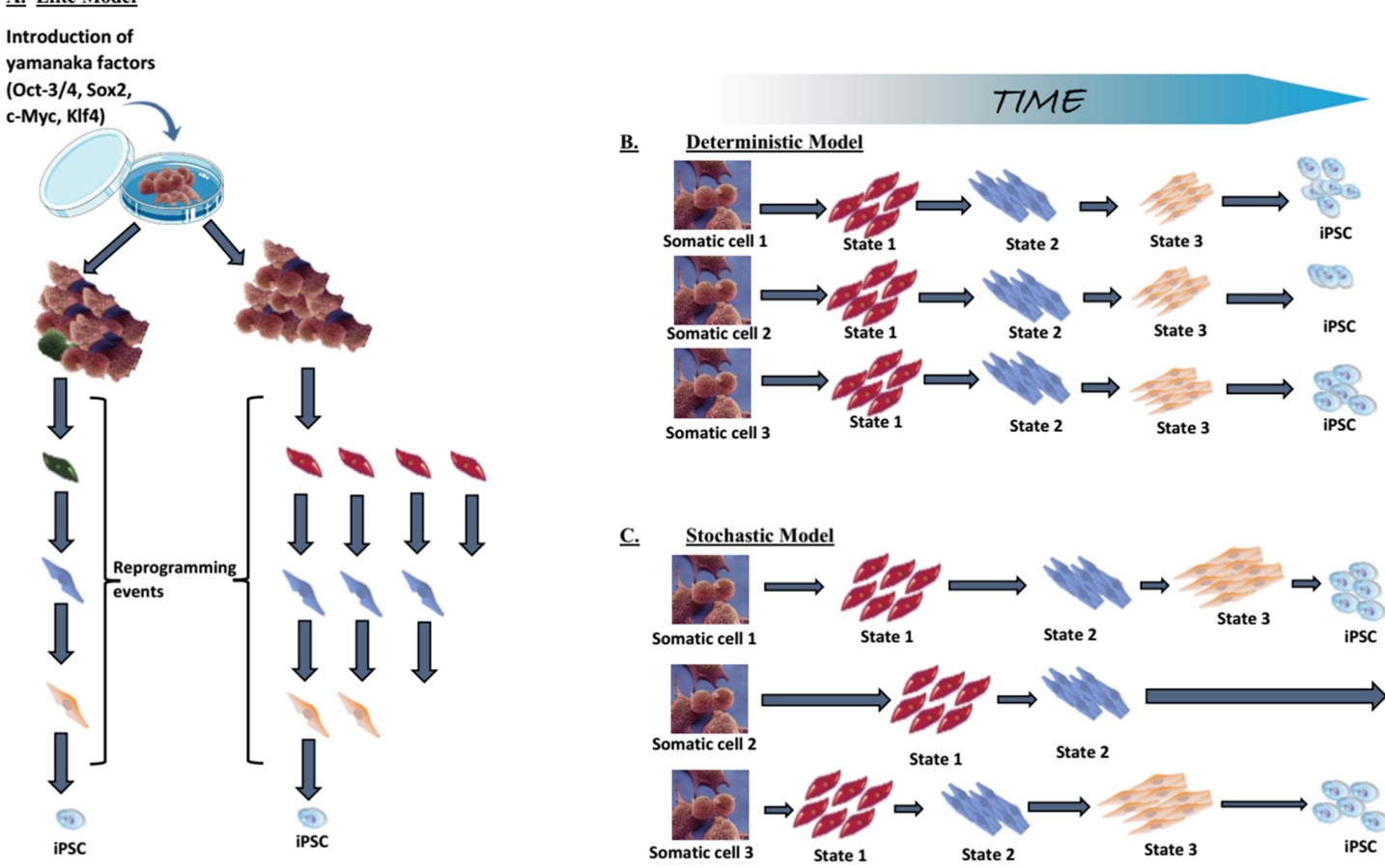

**Figure 5 Mechanistic insights into transcription factor-mediated reprogramming.** (A) The elite model, (B) the deterministic model, and (C) the stochastic model. Adapted from *Takahashi & Yamanaka (2016)*.

earlier in this review. Under the same culture conditions, keratinocytes reprogramme 100 times more efficiently and two times faster than fibroblasts (*Aasen et al., 2008*). Haematopoetic stem cells generate iPSC colonies 300 times more than B and T cells, suggesting that the differentiation status of the donor cell type is important (*Eminli et al., 2009*). Hypoxic culture conditions (5% $O_2$) greatly enhances reprogramming efficiency in mouse and human cells (*Yoshida et al., 2009*). Taken together, donor cell types and culture conditions can modulate reprogramming efficiencies.

## iPSCs VERSUS ESCs

Are iPSCs different from ESCs? Some recent comprehensive studies reveal only a *few* differences in global gene expression and DNA methylation patterns, which were more obvious in early passages of iPSCs (*Bock et al., 2011*; *Guenther et al., 2010*; *Newman & Cooper, 2010*). However, comparison studies with relatively smaller cell clones of iPSCs and ESCs revealed *more* significant differences in either gene expression or DNA methylation patterns (*Chin et al., 2009*; *Marchetto et al., 2009*; *Lister et al., 2011*). Some of the differences were attributed to differential activation of promoters by pluripotency

**Table 5  Advantages and limitations of iPSCs technology.**

| Advantages | Limitations |
|---|---|
| Eliminates ethical issues and religious concerns associated with ESCs use | Efficiency of reprogramming is generally low (*Takahashi & Yamanaka, 2006*; *Takahashi et al., 2007*; *Lowry et al., 2008*; *Huangfu et al., 2008b*) |
| Risk of immune rejection is reduced (*Guha et al., 2013*) | Tumorigenesis (*Okita, Ichisaka & Yamanaka, 2007*) |
| Donor cell is easily and non-invasively obtained, no embryo destruction | Risk of insertional mutagenesis from virus based delivery methods (*Takahashi & Yamanaka, 2006*; *Takahashi et al., 2007*; *Yu et al., 2007*; *Okita, Ichisaka & Yamanaka, 2007*) |
| Accessible to large number of patients, unlike ESCs limited by ethical concerns | Increased chances of development of diseases due to factors used (*Hochedlinger et al., 2005*; *Park et al., 2008b*; *Ghaleb et al., 2005*; *Kuttler & Mai, 2006*) |
| Personalization of treatment with patient-specific stem cells and drugs (*Chun, Byun & Lee, 2011*) | Very early days in this field, more basic research are needed |
| Use for disease modelling-they carry the same disease-causing factor as the patient | Complex and polygenic diseases are difficult to be modeled |
| High-throughput screening for drugs and toxicity prediction (*Wobus & Loser, 2011*; *Choi et al., 2013*) | High costs associated with production and characterization of each cell line |
| Allows for gene targeting and gene editing technology to correct mutations (*Choi et al., 2013*) | Suboptimal standardization (*Pappas & Yang, 2008*). Stringent protocols are still needed |

factors and variables such as the exogenous factor combinations, culture conditions, and delivery methods. Altogether, these studies have conflicting conclusions. Thus the answer to the question raised above is not straightforward. One study revealed a similarity in DNA methylation patterns between the iPSCs and the donor somatic cells, suggesting that iPSCs have a residual epigenetic "memory" marks (*Kim et al., 2010a*, *2010b*). Even among ESC populations, there exist epigenetic heterogeneity and variable differentiation potential (*Martinez et al., 2012*; *Osafune et al., 2008*). Thus, the current consensus is that iPSCs and ESCs are neither identical or distinct, but are overlapping cell populations with genetic and epigenetic differences that reflect their origins. Further experiments are essential to ascertain if these noticeable differences have any impact on the potential therapeutic utility of iPSCs.

Though iPSCs offer many advantages when compared with ESCs, there are some limitations associated with iPSCs as well. Table 5 shows the advantages and limitations of the iPSC technology when compared to ESCs.

## POTENTIAL APPLICATIONS OF iPSCs

The iPSC technology offers the opportunity to generate disease-specific and patient-specific iPSCs for *modeling human diseases, drug development and screening,* and *individualized regenerative cell therapy.* These three concepts are illustrated in Fig. 6 and are discussed in this section.

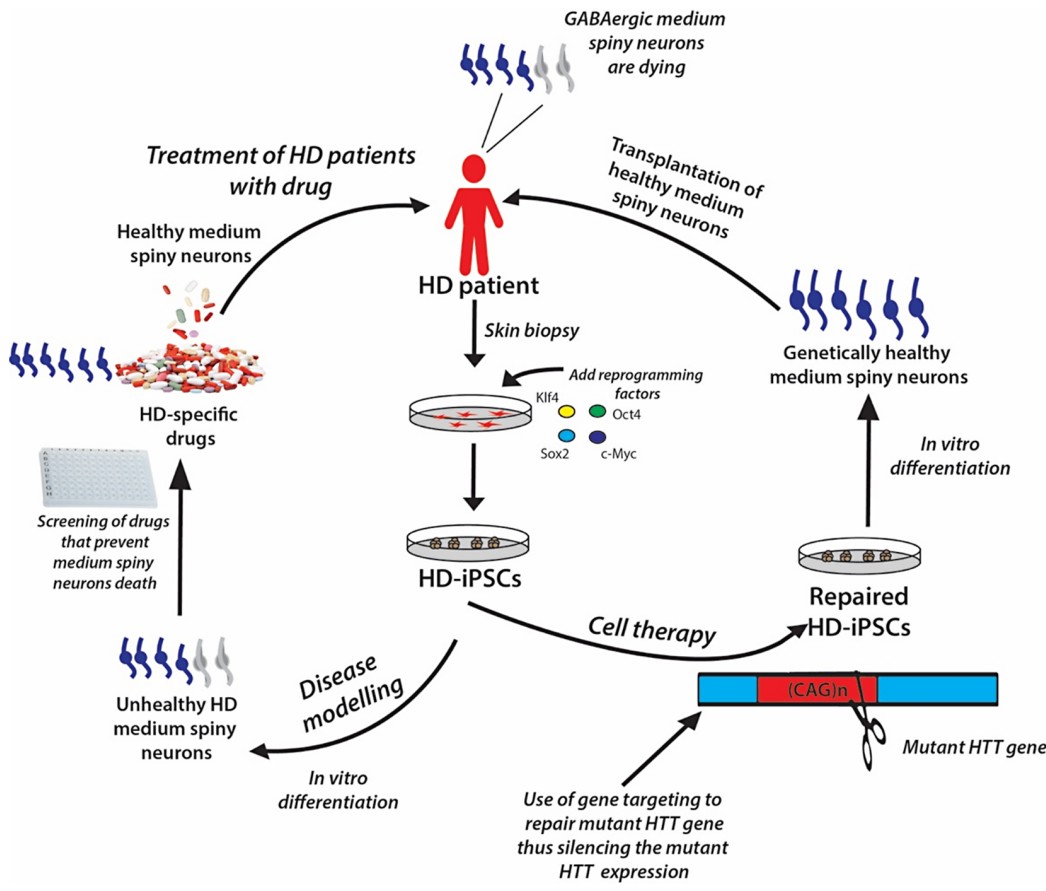

**Figure 6 A schematic showing the potential applications of human iPSC technology for disease modelling, drug discovery and cell therapy using Huntington's disease (HD) as an example.** In HD patients, there is progressive loss of striatal GABAergic medium spiny neurons (MSNs). HD-specific iPSCs generated by cellular reprogramming can be differentiated into striatal MSNs in order to establish an in vitro model of the disease, and potential drugs can be screened leading to discovery of novel drugs that will prevent the degenerative process. Alternatively, if known, the disease-causing mutation (i.e., mutant HTT gene) could be repaired in iPSCs by gene targeting prior to their differentiation into healthy MSNs, followed by transplantation into the patient's brain.

## Disease modeling

Genetically matched iPSC lines can be generated in unlimited quantities from patients afflicted with diseases of known or unknown causes. These cells can be differentiated *in vitro* into the affected cell types, thus recapitulating the "disease in a Petri dish" model. The differentiated, specialized cells or disease models offer the opportunity to gain mechanistic insights into the disease and to use the cells to identify novel disease-specific drugs to treat the disorder; for example, drugs to prevent the death of medium spiny neurons in patients suffering from Huntington's disease (Fig. 6). The ability of iPSCs to proliferate extensively in culture and differentiate into all types of cells in the human body ensures that they can be used as disease models to study many diseases. Certainly, many studies have demonstrated the generation of iPSC lines from patients with various genetically inherited and sporadic diseases (Table 6) (*Wu & Hochedlinger, 2011*). These in vitro studies give the first proof of principle that disease modeling using iPSC technology

**Table 6  Summary of published human iPSC disease models.**

| Disease type | Disease name | Genetic cause | Number of lines | Cell type | Control line | Phenotype | Drug test | PMID |
|---|---|---|---|---|---|---|---|---|
| Neurological | Parkinson's disease | Polygenic | 23 | Dopaminergic neurons | hiPSC | No obvious defect | ND | 19269371 |
| | | Polygenic (with LRRK2 mutation) | 4 | Dopaminergic neurons | hiPSC | Neuronal death with chemical | Yes | 21362567 |
| | Amyotrophic lateral sclerosis | Polygenic | 3 | Motor neurons | hESC | ND | ND | 18669821 |
| | Spinal muscular atrophy | Monogenic | 2 | Motor neurons | hiPSC | Loss of neuron formation, loss of SMN gene expression | Yes | 19098894 |
| | Familial dysautonomia | Monogenic | 2 | Neural crest cells | hiPSC, hESC | Loss of neural crest cells | Yes | 19693009 |
| | RETT syndrome | Monogenic | 4 | Neurons | hiPSC | Loss of synapses, reduced spine density, smaller soma size | Yes | 21074045 |
| | Huntington's disease | Monogenic | 2 | ND | hiPSC, hESC | ND | ND | 18691744 |
| | Friedreich ataxia | Monogenic | 6+ | ND | hESC | Changes GAATTC repeat | ND | 21040903 |
| Blood | Fanconi anemia | Monogenic | 19 | Blood cells | hiPSC, hESC | Corrected loss of FANCA function | ND | 19483674 |
| | Fragile X syndrome | Monogenic | 11 | ND | hiPSC, hESC | Loss of FMR1 expression | ND | 20452313 |
| Cardiac and vascular | Long QT 1 syndrome | Monogenic | 6 | Cardiomyocytes | hiPSC | Increased cardiomyocyte depolarization | Yes | 20660394 |
| | Long QT 2 syndrome | Monogenic | Not reported | Cardiomyocytes | hiPSC | Increased cardiomyocyte depolarization | Yes | 21240260 |
| | LEOPARD syndrome | Monogenic | 6 | Cardiomyocytes | hiPSC, hESC | Increased cardiomyocyte size, decreased MAPK signaling | ND | 20535210 |
| | Timothy syndrome | Monogenic | 16 | Cardiomyocytes | hiPSC | Increased cardiomyocyte depolarization | Yes | 21307850 |
| | Hutchinson Gilford Progeria | Monogenic | 4 | Smooth muscle cells, mesenchyme stem cells | hiPSC, hESC | Smooth muscle and mesenchymal cells apoptosis | ND | 21185252 |
| | | Monogenic | 6 | Smooth muscle cells | hiPSC | Smooth muscle cell nuclear morphology and ageing phenotype | ND | 21346760 |
| | Duchenne muscular dystrophy | Monogenic | 2 | ND | hiPSC, hESC | ND | ND | 18691744 |

| Disease type | Disease name | Genetic cause | Number of lines | Cell type | Control line | Phenotype | Drug test | PMID |
|---|---|---|---|---|---|---|---|---|
| Pancreatic | Type 1 diabetes | Polygenic | 4 | Insulin- and glucagon-producing cells | hESC | ND | ND | 19730998 |
| Hepatic | A1-antitrypsin deficiency | Monogenic | 19 | Hepatocytes | hiPSC | Loss of A1-antitrypsin expression | Yes | 20739751 |
| Others | Prader–Willi syndrome | Monogenic | 4 | Neurons | hiPSC, hESC | Imprint disorder | ND | 20956530 |
| | Angelman and Prader–Willi syndrome | Monogenic | 13 | Neurons | hiPSC, hESC | Loss of paternal UBE3A expression | ND | 20876107 |
| | Down syndrome | Monogenic | 2 | ND | hiPSC, hESC | ND | ND | 18691744 |

**Notes:**
Adapted from *Wu & Hochedlinger (2011)*.
ND, not determined.

is a viable option. However, the aim of disease modeling is to understand the molecular mechanism of diseases, with the ultimate goal of developing drugs for their treatment.

## Drug development and cytotoxicity studies

*Lee et al. (2009)* utilized iPSCs to show disease modeling and drug screening for familial dysautonomia, a rare genetic disorder of the peripheral nervous system (Table 6). The generated familial dysautonomia-iPSCs were screened with multiple compounds, and the authors revealed that a plant hormone, kinetin, can partly normalize the disease phenotype (*Lee et al., 2009*). Loss of neurons following in vitro differentiation of spinal muscular atrophy-iPSCs was ameliorated by exposure to experimental drugs (*Ebert et al., 2009*). These studies and many others (see Table 6) show that iPSCs can facilitate drug screening and discovery. Indeed, several clinical drug candidates have been derived from iPSC studies and are currently in clinical trials (*Bright et al., 2015*; *Naryshkin et al., 2014*; *Mullard, 2015*; *McNeish et al., 2015*). iPSCs are also used for testing for the toxic and non-toxic effect of therapeutic drugs. Itzhaki and colleagues used long QT 2 syndrome cardiomyocytes-iPSCs to test the potency and efficacy of existing and new pharmacological drugs and to assess the cardiotoxic effects and safe dose levels of drugs (*Itzhaki et al., 2011*). As a powerful tool for disease models, drug discovery and cytotoxicity studies, iPSCs offer more advantages over animal models and clinical testing. Animal models do not perfectly mirror the true human disease phenotype, and iPSCs toxicity models are less expensive and save time when compared with conventional testing systems. Additionally, a different response to drug toxicity in animals, due to species differences, could prevent the recapitulation of the full human disease phenotype.

## Regenerative medicine

The iPSC technology offers an exciting opportunity for generating patient-specific stem cells for autologous transplantation. In regenerative medicine, the stem cells are used to promote endogenous regenerative repair or to replace injured tissues after cellular transplantation. The clinical translation of iPSC-based cell therapy is no longer futuristic, as the dream has now been realized. Two ground-breaking preclinical studies provided a proof-of-concept that led to the realization of this dream. In 2007, Jaenisch and colleagues used homologous recombination (gene targeting method) to repair the disease-causing mutations in iPSCs generated from a humanized mouse model of sickle cell anemia (SCA) (*Hanna et al., 2007*). The repaired SCA-iPSCs were differentiated into hematopoietic progenitor cells and subsequently transplanted into the affected transgenic mice. This resulted in the rescue and correction of the disease phenotype. The following year, *Wernig et al. (2008b)* (from Jaenisch's research group) reported an improvement in the dopaminergic function and behavioral symptoms in a rat model of Parkinson's disease, after the transplantation of iPSC-derived dopaminergic neurons. These two successful iPSC-based cell therapies spurred the stem cell research community into exploring iPSCs therapy in humans. The first clinical trial using human iPSC was initiated in 2014 by transplanting human iPSC-derived retinal pigment epithelial (RPE) cells to treat macular degeneration (*Kimbrel & Lanza, 2015*). The progression of the

macular degeneration was halted in the first patient, with improved vision (*Scudellari, 2016*). However, the trial was placed on hold due to the discovery of mutations in the iPSCs of the second patient (*Kimbrel & Lanza, 2015*). The researchers at RIKEN institute are hoping to resume the study using HLA-matched allogeneic iPSCs (*Trounson & DeWitt, 2016*; *Cell Stem Cell Editorial Team, 2016*).

The recent advances in genome editing technology now allow for the introduction of genetic changes into iPSCs in a site-specific manner. We can now repair disease-causing gene mutations in patient-derived iPSCs, thus generating genetically healthy human iPSCs lines for iPSC-based cell therapy (see Fig. 6). Similarly, we can also introduce specific mutations into non-diseased iPSCs, and generate genetically matched isogenic iPSC lines that mimic the true pathology of the disease of interest, to be used for human iPSC-based disease models. Gene editing technologies like *zinc-finger nucleases* (ZFN) (*Hockemeyer et al., 2009*; *Zou et al., 2009*), *transcription activator-like effector nucleases* (*TALENS*) (*Christian et al., 2010*; *Hockemeyer et al., 2011*; *Sanjana et al., 2012*), and *CRISPR-Cas9* (*Cong et al., 2013*; *Perez-Pinera et al., 2013*; *Shalem et al., 2014*; *Jinek et al., 2012*) technology have greatly improved the efficiency of gene editing in both human ESCs and iPSCs via DNA double-stranded breaks at the site of gene alteration. The combination of human iPSC platform with gene editing technologies can make iPSC-based cell therapy a more powerful and viable stem cell therapy option. The following section presents an in-depth analysis regarding gene editing technology in iPSCs generation.

## GENOME EDITING TECHNOLOGY IN iPSCs GENERATION

Induced pluripotent stem cells have been indisputably proven to be a discovery that will transform medicine with respect to understanding the genetic etiology of diseases while equally providing the much needed genetic therapies. Its current combination with genome editing has further enhanced the diagnostic and therapeutic power of the iPSCs (*Hotta & Yamanaka, 2015*). Several methods have been used in the past to genetically target pluripotent stem cells. The process of gene targeting means modifying a specific genomic locus on a host DNA, and the locus is replaced with an exogenous sequence by supplementation with a targeting vector. The technique of gene targeting has availed scientists with the ability to control cellular genomes (*Hotta & Yamanaka, 2015*). Gene targeting has however been shown to be way more challenging in human pluripotent stem cells than in mouse ES cells (*Hotta & Yamanaka, 2015*) and this has been attributed to differences in developmental stages rather than species-related differences (*Shi et al., 2017*). Conventional gene targeting has recorded only a limited amount of success (*Nichols & Smith, 2009*) hence the drive towards developing better methods of gene targeting.

Gene editing technologies have remarkably improved over the years with the recent technologies able to introduce genetic changes in a site-specific manner in iPSCs (*Urbach, Schuldiner & Benvenisty, 2004*). The more recent technologies induce double-stranded DNA breaks in the region of gene modification (*Urbach, Schuldiner & Benvenisty, 2004*). These programmable site-specific nucleases have evolved from

ZFN (*Hockemeyer et al., 2009*; *Zou et al., 2009*) to TALENs (*Hockemeyer et al., 2011*; *Sanjana et al., 2012*) and the RNA guided engineered nucleases (RGEN) gotten from the bacterial clustered regularly interspaced short palindromic repeat (CRISPR)-Cas (CRISPR-associated) nine system (*Perez-Pinera et al., 2013*; *Shalem et al., 2014*). These technologies can easily correct pathology-causing genetic mutations derived from diseased patients and similarly can be used to induce specific mutations in disease-free wild-type iPSCs (*Urbach, Schuldiner & Benvenisty, 2004*). Thus with this approach, genetically matched isogenic iPSCs can be generated while ensuring that true pathologies are reliably identified and not confused with genetic background variations or epiphenomena associated with line-to-line disparities (*Urbach, Schuldiner & Benvenisty, 2004*). In as much as the three nucleases possess a similar mechanism of action which is the cleavage of chromosomal DNA in a location-specific manner, each of the nucleases still has its unique characteristics (*Kim & Kim, 2014*). The well-documented study done by *Kim & Kim (2014)* on the nucleases has been briefly summarized in Table 7. Of the three nucleases, the CRISPR-Cas9 system has however gained wide acceptance and usage in the editing of human iPSC because it is simple to design and use (*Urbach, Schuldiner & Benvenisty, 2004*), thus necessitating a little more review below.

Cas9 is a large multifunctional protein having two putative nuclease domains, the HNH and RuvC-like (*Doudna & Charpentier, 2014*). The HNH and the RuvC-like domains cleave the complementary 20-nucleotide sequence of the crRNA and the DNA strand opposite the complementary strand respectively (*Doudna & Charpentier, 2014*). Several variants of the CRISPR-Cas9 system exists and hence the subtle diversity to their modes of action: (1) The original CRISPR-Cas9 system functions by inducing DNA double-stranded breaks which are triggered by the wild-type Cas9 nuclease directed by a single RNA (*Urbach, Schuldiner & Benvenisty, 2004*). However, its major challenge is the possibility of off-target effects (*Urbach, Schuldiner & Benvenisty, 2004*). (2) The nickase variant of Cas9(D10A mutant) which is generated by the mutation of either the Cas9 HNH or the RuvC-like domain (*Li et al., 2011*; *Christian et al., 2010*) is directed by paired guide RNAs. (3) Engineered nuclease variant of Cas9 with enhanced specificity (eSpCas9) (*Xiao et al., 2013*; *Gupta et al., 2013*). The nickase (D10A mutant) and the eSpCas9 variants have both been shown to substantially reduce off-target effects while still maintaining their meticulous on-target cleavage (*Xiao et al., 2013*; *Gupta et al., 2013*). (4) Catalytically dead Cas9 (dCas9) variant is generated by mutating both domains (HNH and RUvC-like) (*Li et al., 2011*; *Christian et al., 2010*). dCas9, when merged with a transcriptional suppressor or activator can be used to modify transcription of endogenous genes (CRISPRa or CRISPRi) or when fused with fluorescent protein can be used to image genomic loci (*Xiao et al., 2013*; *Gupta et al., 2013*; *Cho et al., 2014*). (5) A modified CRISPR-Cas9 variant has been used to efficiently introduce DNA sequences in an exact monoallelic or biallelic manner (*Gaj, Gersbach & Barbas, 2013*), and (6) CRISPR-Cas9 fused with cytidine deaminase, results in a variant which induces the direct conversion of cytidine to uridine, hence circumventing the DNA double-stranded break (*Segal & Meckler, 2013*).

**Table 7  Summary of the nucleases used in genome editing for iPSCs generation.**

| Nuclease | Composition | Availability | Targetable sites | Pitfalls |
|---|---|---|---|---|
| **(a) ZFN** | ZFN is composed of a modular structure which has two domains: a DNA-binding Zinc-finger protein (ZFP) domain and a nuclease domain gotten from the *FokI* restriction enzyme<br><br>The *FokI* nuclease domain has to dimerize in order to cleave DNA<br><br>ZFPs determines the ZFNs sequence specificity, which comprise of $C_2H_2$ zinc-fingers tandem arrays—the DNA-binding motif that is most common in higher eukaryotes | By modular assembly of pre-characterized zinc-fingers, it is quite convenient to construct new ZFPs with desired specificities<br><br>Available resources for programmable nucleases have been extensively elucidated by *Kim & Kim (2014)* | Sites that can be successfully targeted are often rich in guanines and consists of 5′-GNN-3′ (where N stands for nucleotide) repeat sequences | The ZFNs created through the convenient method of zinc-fingers pre-characterization are often devoid of DNA targeting activity or are often cytotoxic owing to off-target effects<br><br>Constructing ZFNs with high activity and low cytotoxicity still remains a challenge with the use of publicly available resources<br><br>The use of ZFNs are hampered by poor targeting densities<br><br>Presently no available open-source collection of 64 zinc-fingers that can cover all the likely combinations of triplet sites<br><br>Chromosomal DNA cannot be cleaved efficiently by all newly assembled ZFNs, especially those having three zinc-fingers |
| **(b) TALENS** | Although the TALENS use a different category of DNA-binding domains named transcription activator-like effectors (TALEs), they however, still contain the *FokI* nuclease domain at their carboxyl termini | New TALENs with desired sequence specificities can be easily designed because of the one-to-one correspondence between the four bases and the four RVD modules | The crucial advantage of TALENs over the other nucleases is that it can be designed to target almost any desired DNA sequence | The fact that TALENs frequently consists of about 20 RVDs and that highly homologous sequences can fuse with one another in cells, make the construction of DNA segments that encode TALE arrays challenging and time-consuming |

(Continued)

| Nuclease | Composition | Availability | Targetable sites | Pitfalls |
|---|---|---|---|---|
|  | The TALEs are made up of 33–35 amino acid repeats. Repeat variable diresidues (RVDs) determines the nucleotide specificity of each repeat domain. The four different RVDs include: Asn–Ile, His–Asp, Asn–Asn, Asn–Gly—these are most widely used to recognize adenine, cytosine, guanine and thymine respectively | Available resources for programmable nucleases have been extensively elucidated by *Kim & Kim (2014)* | Although conventional TALENs do not cleave target DNA containing methylated cytosine, interestingly, a methylated cytosine is identical to thymine in the major groove. Therefore, Asn–Gly RVD repeat (which recognizes thymines) can be used to replace His–Asp RVD repeat (which recognizes cytosines) and thus generate TALENs that cleave methylated DNA | The need for a thymine to be at the 5′ of the target sequence for recognition by two amino-terminal cryptic repeat folds appear to be the only limitation to the construction of the TALENs |
| **(c) RGEN** | The organisms bacteria and archaea capture small fragment of the DNA (∼20 bp) form the DNA of invading plasmids and phages and fuses these sequences (named protospacers) with their own genome thus forming a CRISPR. For type II CRISPR, the CRISPR sites are first transcribed as pre-CRISPR RNA (pre-crRNA) and further processed to form target-specific CRISPR RNA (crRNA). Also contributing to the processing of the pre-crRNA is the invariable target-independent trans-activating crRNA (tracrRNA), which is also transcribed from the locus. An active DNA endonuclease (termed dualRNA-Cas9) is formed from when Cas9 is complexed with both crRNA and tracrRNA. A single-chain guided RNA can be formed by linking crRNA and tracrRNA, this simplifies the RGEN components | 20 bp guide DNA sequences can be cloned into vectors that encode either crRNA or sgRNA and this easily generates new RGEN plasmids. New RGEN formation does not require complicated protein engineering because Cas9 stays the same. Available resources for programmable nucleases have been extensively elucidated by *Kim & Kim (2014)* | A 23 bp target DNA sequence is cleaved by the formed DNA endonuclease, this target DNA sequence is made up of the 20 bp guide sequence in the crRNA (which is the protospacer) and the 5′-NGG-3′, also 5′-NAG-3′ (but to a lesser degree) a sequence regarded as the protospacer adjacent motif (PAM), recognizable by Cas9 itself. RGENs cleave methylated DNA as opposed to TALENs and ZFNs | The need for a PAM sequence is a limitation for the RGEN target sites. The need for guanine to be at the 5′ end is also another limitation for the targetable sites as RNA polymerase III transcribes guide RNAs under the guidance of the U6 promoter in cells. RGENs in cells do not efficiently cleave all sequences that contain the PAM sequence |

Hotta and Yamanaka have extensively reviewed how these nucleases have been used to mediate gene editing in pluripotent stem cells (*Hotta & Yamanaka, 2015*). Thus it is anticipated that the combination of these two technologies (gene editing and iPSCs) might be the dawn of a new phase of gene therapy.

## FUTURE PERSPECTIVE

The promise that iPSCs are viable and possibly superior substitutes for ESCs in disease modeling, drug discovery and regenerative medicine has not yet been fulfilled. Despite great success in animal models, there are still many obstacles on the road to the clinical application of iPSCs. A major limitation is the heterogeneous nature of the cell population and differentiation potential of iPSCs. Hopefully, the *CRISPR-Cas9* system can be used to address this limitation since the technology can improve the disease phenotype of differentiated cells (*Hotta & Yamanaka, 2015*; *Deleidi & Yu, 2016*). Another major limitation is the lack of robust lineage-specific differentiation protocols to generate large quantities of purified and matured iPSC-differentiated cells. More basic research on reprogramming technology is critical for the development of novel protocols for the generation of standardized human iPSC. A more current biotechnology, the microRNA switch (*Miki et al., 2015*), is expected to facilitate the maturation and purification of iPSC-differentiated cells and to reduce clonal variation.

While we wait for these limitations to be addressed, it will be wise to bank iPSCs from patients with specific diseases. Doing so will allow us the time to guarantee the quality of these cells thus saving time and cost when they are made available for transplantation.

## CONCLUSION

The discovery of iPSCs by Takahashi and Yamanaka is truly a breakthrough of the decade in stem cell science. The year 2016 marked the 10th anniversary of this landmark discovery. The last decade has witnessed remarkable advancement in our understanding of the molecular mechanisms of induced pluripotency, and we moved from the "bench to the bedside" in 2014. The more recent long-term study involving the application of human iPSC-derived dopaminergic neurons in primate Parkinson's disease (PD) models at the Center for iPS Cell Research and Application, Kyoto University, Japan, reveals that human iPSCs are clinically applicable for the treatment of patients with PD (*Kikuchi et al., 2017*). The iPSC-based cell therapy is still at its infant stage. The remaining barriers blocking the path to successful translation of this technology into clinical therapy have to be overcome. We believe many of these challenges are only technical and with time "*this too shall pass away*." The combination of the human iPSC technology with genome-editing technologies may trigger a new era of gene therapy utilizing iPSCs.

## ACKNOWLEDGEMENTS

The author wish to thank Kingsley Nnawuba and David Adeiza Otohinoyi for their assistance with some of the figures and tables.

### Funding

The authors received no funding for this work.

### Competing Interests

The authors declare that they have no competing interests.

### Author Contributions

- Adekunle Ebenezer Omole prepared figures and/or tables, authored or reviewed drafts of the paper.
- Adegbenro Omotuyi John Fakoya prepared figures and/or tables, authored or reviewed drafts of the paper.

### Data Availability

There is no raw data for this literature review.

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
