# Peer review of "Ten years of progress and promise of induced pluripotent stem cells: historical origins, characteristics, mechanisms, limitations, and potential applications"

_PeerJ, doi:10.7717/peerj.4370_

## Round 0.1 · original submission · Major Revisions

The review covers a broad and growing amount of publications including survey of recent reviews. It needs to be improved with a high number of minor revisions and grammatical corrections that reviewer 1 has listed in the attached PDF. The figures and tables adapted from previous published reviews are adequately quoting the original sources. As some of these reviews are not accessible under open access (in particular reference 117 noted by reviewer 2), the adapted figures proposed here provide a useful synthesis for an easy reading and thus should be kept as well as the tables, but reference 82 (line 1011) must be corrected for their authors: "Brouwer M, Zhou H, and Nadif Kasri N." (and not " ... and Kasri, N.N.").

A careful implementation of all corrections and clarifications listed by both reviewers is necessary and will improve the readability of this review.

Reviewer 1 ·

Basic reporting

This manuscript reviews the advances in the field of induced pluripotent stem cells (iPSC) during the 11 years since iPSC were first described. Overall the review is detailed with appropriate citations, over 200, and the organization into different sections is well thought out. Considering the growth of research in this field, it is difficult to maintain this type of review up to date although the authors have done well. Keeping in mind that several other similar reviews on the subject have been published, this work represents a considerable body of knowledge useful both to newcomers to the field and for experienced authors.

Unfortunately, the manuscript contains a significant number of grammatical errors throughout the text, particularly concerning pluralities and tense. For the authors help, I have included a list of potential corrections and suggestions for several phrases where the meaning was unclear.

Experimental design

The manuscript is clearly in the scope of PeerJ, with well-outlined objectives. Even though the field is expanding continually, the analysis performed by the authors is overall of appropriate depth, detail and rigorous.

Major point:
While through most of the manuscript, the authors provide a well detailed description of the many of the advances in the field, page 6 line 167, the manuscript would be strengthened by including a deeper description of the methods enabling development of fully reprogrammed iPSC.

Validity of the findings

In the concluding sections, the authors provide a very appropriate assessment of the future potential of iPSC and the barriers we face before effective clinical deployment.

Additional comments

Clearly, the manuscript was written last year, 2016, which may account for many of the problems with tense.

Minor points:
Pg:8 line 241 The full names of DOT1L MBD3 etc. need to be provided.
Pg:12, line 344, “highly immunogenic.” needs a reference.
There are 9 possible loops in the Oct4/Sox2/Nanog transcription machinery, so while I believe the term auto regulatory loop can be used to describe this circuitry overall, in the detailed description of this mechanism, it would be more appropriate to refer to loops.
Pg: 14-16 the authors make use of both the terms iPSCs and PSCs and it is not clear if the latter is being used to describe pluripotent stem cells as a whole.

Reviewer 2 ·

Basic reporting

Please see General comments below

Experimental design

Please see General comments below

Validity of the findings

Please see General comments below

Additional comments

This is a very broad review of various aspects of iPSCs. It adequately covers many of the major issues relating to this technology, but because of its broad scope, lacks synthesis and critical evaluation of some of the cited work. The figures and tables are mostly adapted from previously published reviews and do not incorporate more recent work. I would recommend major revisions be carried out before the mansucripts publication.

Please see below for specific comments on the manuscript.

Line 136 : "The search started in year 2000 leading to a list of 24 candidate reprogramming factors" - This statement refers to the paper published in Cell in 2006 not 2000.

Lin 289 : "self-deleting peptide" - more common usage for this would be "self-cleaving peptide"

Instead of separating each delivery methodology to its own paragraph they should be grouped under two major headings (integrating vs nonintegrating)

Secondary IPSC systems (knock-in mouse models of Trx Factors) are not considered adequately - especially when disccussing the factors that limit the efficiency of reprogramming (Section 5.8).

Section 5.6 seems to dwell on the same topic as 5.1 and could be merged to make the text clearer

The authors use "we" in several places in the text (eg. Line 688). Since the work they cite is not theirs per se, it would be more appropriate to use "one" or the passive voice.

Section on the details of Cas9 (line727-743) seems out of place in the context of this iPSC review.

Why is "microrna switch" on line 772 italicized?

Line 787- Should read as "believe"

Many of the figures are "adapted" from reference 117 - The authors have not really added onto these, so I wonder if they are really necessary. One can also read teh origival review

Simple adapting a table (Table 2 and 4) from an older review is not acceptable in my opinion. More recent information has to be added on these tables.

---

## Round 0.2 · accepted · Accept

Although a high number of small points needed to be addressed thus qualifying the revision required as major, this revised manuscript has been carefully corrected throughout and the suggestions by both reviewers adequately answered. Therefore the manuscript is now acceptable for publication.